# Seroepidemiology of *Borrelia burgdorferi* s.l. among German National Cohort (NAKO) Participants, Hanover

**DOI:** 10.3390/microorganisms10112286

**Published:** 2022-11-17

**Authors:** Max J. Hassenstein, Irina Janzen, Gérard Krause, Manuela Harries, Vanessa Melhorn, Tobias Kerrinnes, Yvonne Kemmling, Stefanie Castell

**Affiliations:** 1Department of Epidemiology, Helmholtz Centre for Infection Research (HZI), 38124 Braunschweig, Germany; 2PhD Programme “Epidemiology” Braunschweig-Hannover, Germany; 3German Center for Infection Research (DZIF), Braunschweig, Germany; 4Hanover Medical School (MHH), 30625 Hannover, Germany; 5TWINCORE, Centre for Experimental and Clinical Infection Research, A Joint Venture of the Hannover Medical School and Helmholtz Centre for Infection Research, 30625 Hannover, Germany; 6Department of RNA-Biology of Bacterial Infections, Helmholtz Institute for RNA-Based Infection Research, 97080 Würzburg, Germany

**Keywords:** borrelia burgdorferi, lyme disease, tick-borne diseases, seroepidemiologic studies, German National Cohort, force of infection, borrelia infections

## Abstract

Lyme borreliosis is the leading tick-related illness in Europe, caused by *Borrelia Burgdorferi* s.l. Lower Saxony, Germany, including its capital, Hanover, has a higher proportion of infected ticks than central European countries, justifying a research focus on the potential human consequences. The current knowledge gap on human incident infections, particularly in Western Germany, demands serological insights, especially regarding a potentially changing climate-related tick abundance and activity. We determined the immunoglobulin G (IgG) and immunoglobulin M (IgM) serostatuses for 8009 German National Cohort (NAKO) participants from Hanover, examined in 2014–2018. We used an enzyme-linked immunosorbent assay (ELISA) as the screening and a line immunoblot as confirmation for the *Borrelia Burgdorferi* s.l. antibodies. We weighted the seropositivity proportions to estimate general population seropositivity and estimated the force of infection (FOI). Using logistic regression, we investigated risk factors for seropositivity. Seropositivity was 3.0% (IgG) and 2.1% (IgM). The FOI varied with age, sharply increasing in participants aged ≥40 years. We confirmed advancing age and male sex as risk factors. We reported reduced odds for seropositivity with increasing body mass index and depressive symptomatology, respectively, pointing to an impact of lifestyle-related behaviors. The local proportion of seropositive individuals is comparable to previous estimates for northern Germany, indicating a steady seroprevalence.

## 1. Introduction

Lyme borreliosis (LB), also known as Lyme disease, is the most common tick-borne disease in Europe, with *Ixodes ricinus* as the predominant tick species [1,2]. Spirochaete bacteria of the *Borrelia burgdorferi* sensu lato (*B. burgdorferi* s.l.)—a complex transmitted from ticks to humans—can cause multisystem disease with several states of manifestation. In the early stages of the infection, erythema migrans, a skin rash circling the tick bite, is the most frequent symptom. Other clinical manifestations at advanced stages are neuroborreliosis or Lyme arthritis [3]. However, 5% of tick-bitten humans seroconvert, and 2% of the tick-bitten develop clinical disease [4].

In Germany, LB is a notifiable infection in nine of sixteen federal states, covering eastern Germany, Bavaria, Rhineland-Palatinate, and Saarland as the only German states in the western region; this hampers nationwide monitoring of cases for risk assessment and subsequent public health measures [5]. In the nine states with mandatory notification, the annual mean incidence from 2013 to 2017 was 33 notified cases per 100,000 persons, with a seasonal spike from June to August and a stable incidence over the years [6]. Akmatov et al. reported 429 LB diagnoses per 100,000 insured persons in 2019, based on recent health insurance data [7]. Both notification and health insurance data indicate regional differences in disease burden caused by LB. This heterogeneity is also supported by monitoring data, suggesting a variation in tick density, distribution, and infection status with *Borrelia* spp. within Germany [8,9,10,11]. Lower Saxony, a northern German federal state, seems to have a higher average proportion of infected ticks (30.6%) than central European countries taken together (19.3%) [8,12]. For Hanover, Lower Saxony, in 2018 and 2019, the proportion of *Borrelia* spp.-infected ticks was 31.1% in mixed forests, 32.9% in urban areas, and 35.5% in broadleaved forests [8]. Retrospectively, Hanover’s proportion of infected ticks remained constant from 2005 to 2015 [13,14]. However, the tick density seems to have increased between 2017 and 2018 [15]. Because of climate change, the tick season will potentially extend from currently March to October [6,16] to include the fall and winter months due to higher weekly mean temperatures [17], enabling tick activity if above 7 °C [18]. Considering Lower Saxony, the average yearly temperature has increased in the last decades from 8.6 °C in 1961–1990 to 9.3 °C in 1981–2010 [19]. In addition, the likelihood of human exposure might elevate due to more outdoor activities with rising air temperatures [20].

Previous research considering the notification and serological data suggests several risk factors for human seropositivity, with advancing age and male sex as influential factors [21,22,23,24]. However, surveillance and health insurance data display a more complex representation of the age distribution: a bimodal distribution of cases over age with an elevated incidence and diagnoses in children aged 5 to 9 years and adults aged 50 to 69 [6,16,25]. Compared with females, males are more affected in childhood and less in adulthood. Furthermore, existing research has not yet clarified the role of socioeconomic status (SES) conclusively, as the studies use a different methodology. For example, the use of income on the neighborhood/municipality level versus the use of education level on the individual level found differing results [24,26,27,28,29,30]. The interrelationship between SES, living environment, profession, and lifestyle, including recreational activity, may affect tick exposure, and, therefore, past infections are reflected in part by the serostatus.

We derived three epidemiological objectives in order to fill the existing research gaps: First, we aimed to estimate *B. burgdorferi* s.l. seropositivity in the general adult population of Hanover—a city with no active public health surveillance of LB or recent serosurveys but with abundant tick data, as mentioned above. Here, a comprehensive serological study can offer insights into local seropositivity and a baseline for future serosurveys, considering potential climate-related exposure shifts. Second, we aimed to estimate the serology-based force of infection (FOI), the rate at which the susceptible acquire an infection [31] with *B. burgdorferi* s.l., to provide a different perspective on the age distribution and risk of seropositivity. Finally, we used extensive German National Cohort (NAKO) data to evaluate the acknowledged and debated risk factors.

## 2. Materials and Methods

### 2.1. Study Sample

We investigated data from 8009 German National Cohort (NAKO) participants examined at the study center in Hanover between October 2014 and November 2018 [32]. The study center invited people aged 20 to 69 years randomly drawn into 10-year age groups from population registries with primary residences in Hanover. We oversampled the age groups 40 and above. After providing written informed consent, the participants underwent comprehensive standardized examinations, including health-related interviews, self-administered questionnaires, physical and medical examinations, and provided biomaterials, including a 65 mL blood sample per participant drawn by a study assistant. The sample collection and subsequent processing were subject to strict standardization and took a maximum period of 2 h. All samples were aliquoted and frozen at −80 °C at the Hanover Unified Biobank, located within the same building. Eligibility for study participation was irrespective of their health condition, as long as the invited participant was able to provide written informed consent, reach the study center independently, and participate in most examinations. For travel expense compensation, study personnel offered EUR 10 to each participant at the end of the examinations.

### 2.2. Blood Sample Analyses

A DIN EN ISO (German Institute for Standardization/International Standardization Organization/European standard) 15,189 accredited and ISO 9001 certified contract laboratory conducted the serological analyses of blood samples for immunoglobulin G (IgG) and immunoglobulin M (IgM) antibodies against *B. burgdorferi* s.l. on our behalf. For transport to the external laboratory in July 2020, the samples were packaged with dry ice, and temperature logs ensured an adequate temperature during transportation. The samples were stored at −20 °C at the laboratory before processing. The laboratory performed a two-tier antibody testing procedure consisting of an enzyme-linked immunosorbent assay (ELISA) as a screening test and a line blot immunoassay (line blot) as a confirmatory test in case of positive or equivocal ELISA results, in line with the current microbiologic-infectiologic quality standards (MiQ12) [33].

For the ELISA screening test, our contract laboratory used the “*Borrelia afzelii* and VlsE IgG Europe ELISA” and “*Borrelia afzelii* IgM ELISA” kits, Virotech Diagnostics GmbH, with a >99% sensitivity (IgG and IgM) and 97% (IgG) or 98.8% (IgM) specificity, respectively. The antigens covered by these kits were the *B. burgdorferi* strain ZS7, *B. garinii* strain PBr, and *B. afzelii* strain Pko.

The screening test returned a Virotech Unit (VU) value, an arbitrary antibody quantification scale used for the initial classification of the samples as positive (VU > 11), equivocal (VU ≤ 11 and VU ≥ 9), or negative (VU < 9). Then, as a confirmatory test (line blot) for the positive or equivocal ELISA results, our contract laboratory used the “WE225 *Borrelia* Europe plus TpN17 LINE IgG” and “WE224 *Borrelia* Europe LINE IgM” kits, Virotech Diagnostics GmbH, with a >99.9% sensitivity and 98% specificity (IgG and IgM). The antigen strains considered by these kits were the OpsC (p23) from *B. afzelii*, VlsE recombinant from *B. burgdorferi* B31, p39 (BmpA) recombinant from *B. afzelii* PKo, DbpA (Pko) and DbpA (PBi, PBr, A14 S) from B. *bavariensis* PBi and *B. garinii* PBr, p58 (OppA-2) recombinant from *B. bavariensis* PBi, and p83/100 recombinant from *B. afzelii* PKo. In addition, the kits considered the strain EBV VCA-gp125 (affinity purified) for the exclusion diagnostics of Epstein–Barr virus.

### 2.3. Defining B. burgdorferi s.l. Seropositivity

We determined the IgG and IgM serostatuses for the *B. burgdorferi* s.l. antibodies by applying three seropositivity algorithms independently:A positive or equivocal screening test (ELISA) with a subsequent positive confirmatory test (line blot), which corresponds to the current standard MiQ12 [33];A positive screening test (ELISA) with a subsequent positive or equivocal confirmatory test (line blot) or equivocal screening test with a positive confirmatory test; also applied, e.g., in [22,23];A positive screening test (ELISA); also applied, e.g., in [34,35].

In our work and analyses, we primarily use the definition corresponding to the MiQ12 standard (definition 1) and report other definitions for comparability to relevant studies from the literature. To estimate the seropositivity of the local general population in Hanover, we weighted our sample with regard to the local age and sex distribution from the 2020 cohort-component-based population update of the 2011 census (obtained from www.destatis.de and accessed on 6 May 2022) [36] using the iterative proportional fitting (raking) method employing the R-package “survey” [37,38].

### 2.4. Force of Infection

We estimated the serology-based force of infection (FOI) to gain an additional viewpoint on age distribution and infection risk. In our approach, the FOI reflects the balance of seroconversion and seroreversion, reflecting the average change in the seropositivity proportion of the population.

We considered those subjects aged 20–69 years and focused on IgG seropositivity. Our estimation utilizes definition 1, i.e., the detection of antibodies against *B. burgdorferi* s.l., corresponding to the MiQ 12 standard (Section 2.3., definition 1). We did not proceed with an FOI calculation for the IgM class antibodies as a short-term marker for infection due to the non-distinct duration of serum traceability [39], the low crude proportion of IgM positives, and no indication of varying IgM seropositivity over age, as reported before [24]. For the FOI calculation, we considered three models: Muench’s catalytic model [40,41] with a constant FOI over age; Griffiths’ model [42], allowing for a linear FOI increase with age; and Grenfell and Anderson’s model [43] with polynomial functions, allowing for a varying FOI over age. We implemented all three models in the framework of generalized linear models (GLM) (Appendix A), as suggested by Hens et al. [31]. We compared the FOI models regarding the Akaike information criterion (AIC) to determine the best-fitting model.

### 2.5. Regression Analysis

We constructed two binary logistic regression models to obtain the odds ratios (OR) for IgG (model 1) and IgM (model 2) seropositivity for *B. burgdorferi* s.l. as a function of age, sex, migration background, education, net equivalent monthly income, body mass index (BMI), a depression score and smoking status. For the migration background, we used a set of indicators that considered the participant’s residence post-birth, spoken native language, and the subjects’ parents’ country of birth, as suggested by Schenk [44]. Furthermore, we defined the self-reported education levels as low, intermediate, and high, corresponding to the International Standard Classification of Education level (ISCED97) [45]. We used the International Obesity Task Force classification for BMI [46] and assessed the self-reported depression severity in line with the 9-question Patient Health Questionnaire (PHQ-9) [47]. We included smoking status as a proxy for lifestyle [48].

We imputed missing values among our independent variables using multivariate imputation by chained equations (MICE) using the R-package “mice” [49,50]. We adjusted the algorithm to create 73 imputations according to the proportion of incomplete cases, with a maximum of ten iterations per imputation [51,52] with predictive mean matching for the missing numeric values. To ensure adequate missing value imputation, we compared the distributions of the original variable with the imputed variables, considering the mean, standard deviations, interquartile range, and graphical comparison of the imputations with the original data and found no relevant deviation.

We tested the assumption of linearity between the independent continuous variables and the log odds of the outcome variable, employing the Box-Tidwell test. As the test indicated linearity violations, we considered fractional polynomial (FP) functions [53], which provide flexible parameterization to improve a model’s fit. We employed an automated multivariable fractional polynomial (MFP) procedure [54], iteratively cycling through the FP transformations to identify the most suitable FP function for the respective continuous variable. In all instances, the residual deviance did not improve; therefore, we did not consider FP in our final models. Furthermore, we considered two interaction terms in our final model based on the literature [6,21] and exploratory data analysis: age and sex, as well as age and BMI, as BMI linearly increased with advancing age in our sample. In our model for IgG serostatus, both interaction terms were not statistically significant; therefore, we assumed no considerable interaction effect. However, the age and sex interactions were significant in our IgM serostatus. To control for multicollinearity, we calculated the variance inflation factor (VIF) [55]; VIF values above five indicate considerable multicollinearity [56]. We conducted all data wrangling, analyses, and visualization in RStudio (version 2022.02.3, Build 492; R-base, version 4.1.2 [57]).

## 3. Results

We analyzed 8009 participants who provided blood samples that were available for our investigation (Table 1). The participants’ median age was 50 years (interquartile range [IQR]: 42–60), with a maximum age of 74 years. The sample comprised 50.2% females, and 79.8% reported no migration background; 55.9% of the participants reported a high education, and 33.5% reported a medium education. The monthly net equivalent income was distributed relatively evenly across the quartiles; 44.7% and 35.1% of individuals had a normal or pre-obesity body mass index (BMI), respectively. Most subjects reported no/minimal (61.4%) or mild (22.2%) depressive symptoms. Information on smoking status was available for 4178 subjects: 21.2% were never smokers, and 17.4% were former smokers.

All 8009 samples underwent the two-tier antibody testing procedure for IgG and IgM antibody presence for *B. burgdorferi* s.l. with ELISA used as the screening test and a line blot as the confirmatory test (Figure 1, Appendix A). Screening for IgG, 564 (7.0%; 95% CI 6.4–7.6%) tested positive and 390 equivocal (4.9%; 95% CI 4.4–5.4%), of which 252 (26.4%; 95% CI 23.6–29.2%) were confirmed positive, resulting in a 3.1% (95% CI 2.7–3.5%) crude IgG seropositivity (Table 2). Considering IgM testing, 160 screened positive (2.1%; 95% CI 1.8–2.4) and 124 (1.6%; 95% CI 1.3–1.9%) equivocal, of which 76 (26.8%; 95% CI 19.0–34.6%) tested positive in the confirmatory test, resulting in a 0.9% (95% CI 0.7–1.1%) crude IgM seropositivity. Of all 8009 samples, 15 (0.19%; 95% CI 0.1–0.31%) tested positive for both the IgG and IgM antibodies.

Applying the three seropositivity definitions yielded varying proportions of seropositivity (Table 2). Comparing the most stringent (MiQ12) with the least stringent algorithm (ELISA only), the proportion of seropositive samples was more than doubled for both antigens: 3.1% vs. 7.0% (IgG) and 0.9% vs. 2.0% (IgM). The weighted estimates for the local seropositivity among the Hanoverian general population are close to the crude proportions, with slight fluctuations in the decimal place. For Hanover, the local IgG seropositivity is estimated at 3.0% (95% CI 2.7–3.4), and for IgM, at 2.1% (95% CI 1.8–2.4).

Among the three models for the FOI estimation, Grenfell and Anderson’s model with polynomial functions performed best in terms of the AIC value compared to Muench’s and Griffith’s model (Appendix A). In our estimation, the FOI represents the annual average change in the population’s seropositivity proportion and, therefore, a mix of seroconversion and seroreversion. Muench’s constant model estimated the FOI at 0.000637 for all ages (Appendix A). Griffiths’ FOI model estimated 0.000634 for individuals aged 20–24 years and 0.000664 for the highest age group, 65–69 years. Grenfell and Anderson’s model estimated varying FOIs over ages, with 0.0000656 for the participants aged 20–24, then negative estimates for the subjects aged 25–39 years, then 0.0000375 for the 40–44 year-olds, increasing to 0.00317 in the participants aged 65–69 years. The predicted seropositivity from Grenfell and Anderson’s model lies close to the observed mean values across 10-year age groups (Figure 2), except for the age group 30–39, where seropositivity prediction lies comparably far from the observed mean probably due to data dispersion within the data of this age group. Overall, predicted seropositivity lies within the 95% confidence interval (CI) of the observed values, and both the observed and predicted seropositivity show a positive trend with advancing age, with a slight decline in the age group 30–39, when considering the mean observed values; however, 95% confidence intervals (CI) overlap.

From the regression analysis, we found that while holding all other variables constant, the odds for IgG seropositivity increased by 26% (95% CI 13–42%) for every 10-year increase in age (Figure 3, Appendix A). Compared to the females, males had 2.58 times (95% CI 1.94–3.46) the odds for a positive IgG test result. In addition, every point increase in BMI led to a 4% (95% CI 1–7%) reduction in odds for positive IgG serostatus. Comparably, each point increase on the depression scale led to a 6% (95% CI 2–10%) reduction in odds for IgG seropositivity. We found a significant interaction between age and sex for positive IgM serostatus (interaction model not shown): males had 1.57 (95% CI 1.12–2.23) times the odds for positive IgM serostatus for every 10-year increase in age compared to females. At the age of 50, the male sex effect is OR 2.09, and for the age of 70, it is OR 3.00.

We found no indication of multicollinearity in both models [56], as the VIF remained below 1.5.

## 4. Discussion

We have conducted the largest single-site serological survey for antibodies against *B. burgdorferi* s.l. in Germany. Our findings offer a valuable addition to the literature by providing population estimates of local seropositivity, which researchers may use as a baseline reference for future serosurveys in light of potential climate change-related shifts in human tick exposure. We estimated Hanover’s weighted local seropositivity proportion at 3.0% (IgG) and 0.9% (IgM). Our findings confirm advancing age and male sex as risk factors for seropositivity. In addition, we are the first to report that both decreasing BMI and self-reported depression symptomatology are independently associated with positive IgG serostatus. We applied three classification definitions to determine seropositive blood samples to facilitate comparisons with (inter-) national serosurveys, as researchers have been using different definitions [22,24,58], which considerably affect the reported proportion of seropositivity [24], which we also demonstrated in our work.

### 4.1. Hanoverian Seropositivity in Context

For Hanover, we estimated a higher proportion of weighted IgG seropositivity (3.0%) compared with the 2018–2020 estimates for Bonn, North Rhine-Westphalia (NRW), Germany (2.2%, weighted) [24] and elevated IgM seropositivity (0.9% [95% CI 0.7–1.2%] vs. 0.6% [95% CI 0.3–0.8%]), but with overlapping CI. The region around Bonn is deemed to have an increasing tick density and infection [59,60], together with climatic changes promoting tick activity [61], similar to Hanover. However, the Bonn study included only subjects living in the urban core, which may have resulted in lower seropositivity due to lower exposure to ticks. Whereas, in Hanover, we recruited individuals from suburban city areas as well. From the literature, we conclude that federal states with lower levels of urbanization generally have higher proportions of seropositivity, suggesting that individuals living in rural areas are generally more exposed than urbanized individuals are, supported by higher chances of seropositivity among smaller municipalities vs. large municipalities [22,62]. Thus, the reduced tick exposure of Bonn (urbanization), compared to Hanover, may have led to increased seropositivity in Hanover.

Earlier results from nationwide population-based studies, such as the *German National Health Interview and Examination Survey 1998* (BGS98) and the *German Health Interview and Examination Survey for Adults* (DEGS), reported seropositivity for Lower Saxony pooled with other states, combined into northern states. Here, the reported seropositivity for 1997–1999 (BGS: 7.4% [95% CI 5.1–9.6%]) is comparable with our results for Hanover (5.4% [95% CI 4.9–5.6%]) but slightly decreased compared with 2008–2011 (DEGS: 9.1% [95% CI 6.7–11.43%]). Compared with the aforementioned studies, seropositivity also remained constant for Bonn compared with earlier estimates for NRW [23]. When jointly considering these results, we found no increased seropositivity for both regions over the last two decades, despite evidence for increasing density (Bonn, [59]) and locally high but steady tick infection populations (Hannover, [8,14]). Consequently, we conclude that temporal and regional increases in exposure risk, given the evident increases in tick density and abundance as well as the proportion of infected ticks, do not necessarily result in prompt detectable increases in seropositivity, implying that changes in tick populations may not be suited as a direct indicator of human infection and the resulting burden of disease.

However, we advise caution when comparing serosurveys from different locations with no standardized population, as the age and sex distributions may differ, complicating comparability, since both these factors considerably influence seropositivity.

### 4.2. Age-Specific Seropositivity

We found a contrasting age distribution of our serological profile compared with the notified cases. A critical difference is that *B. burgdorferi* s.l. seropositivity underlies varying periods of detectability, given seroconversion and seroreversion, but also lifetime prevalence, while a case of Lyme disease is recorded once by the notification system. Our findings indicate a relatively steady proportion of seropositive samples over the age groups from 20 to 49, with a slight mean seropositivity decrease in the age group 30–39. Generally, our observed age distribution aligns with the previous serological findings from DEGS [22,63] and BGS98 [23,64]. BGS98 also reported a similar seropositivity decrease in subjects aged 30–39, pointing towards an actual age effect and not a cohort effect. In contrast, the notified cases of LB show a bimodal age distribution with local incidence maxima at approximately 5–9 and 60–69 years and its lowest points at 15–19 years [6,16,25,65,66]. Although investigations of seropositivity among children and adolescents are scarce in Germany, one study found a generally increasing trend with age, with the highest odds in adolescents aged 14–17 years compared with children aged 3–6 years [21]. When pooling the serological results from children, adolescents, and adults, seropositivity showed a generally increasing trend with age, with an increasing magnitude of participants aged around 50 years and above, with no bimodal representation of seropositivity across age [22]. The notification data indicate a decrease in incident cases in individuals from approximately 70 years onwards, whereas seropositivity increased in the serosurveys.

Various factors may explain the different age distributions: the notification system records new symptomatic clinical cases of LB [67], i.e., incidence, whereas serosurveys especially report the serostatus, which solely indicates a previous infection and subsequent antibody seroconversion, irrespective of past or present clinical disease. Most symptomatic infections occur in a close temporal relationship to the infection, indicating that the notification data may reflect the risk of new infections by age more accurately than serostatus. Whereas, given the long-lasting detectability of antibodies [39], serosurveys may mainly represent a cumulative effect of lifetime exposure, characterized by the increase in higher age groups [22,23,24], also evident from our findings. Here, decreasing seroreversion (waning antibodies) with age [23,68] can potentially reinforce the cumulative representation of lifetime exposure. The mean seropositivity decrease among individuals aged 30–39, evident from our findings and BGS98 [23], may be due to seroreversion and a lower risk of infection in individuals aged 10–29, which we conclude with caution from a comparatively low number of notified incident cases in this age group [6,16].

We estimated the FOI for IgG seropositivity to gain an alternative perspective on the age distribution. The predicted seropositivity from Grenfell and Anderson’s FOI model offered a reasonable approximation of observed seropositivity over age, indicating an adequate model. The slight decrease in mean seropositivity induced negative mean FOI estimates among individuals aged 25–49 years, which distinctly increased after that, fitting the course of observed seropositivity over age. The negative FOI estimates match our presumption that, on a population scale, individuals aged around 30–39 years may have an elevated ratio of seroreversion to seroconversion, leading to a seemingly declined proportion of seropositivity in this age group.

### 4.3. Risk Factors for Seropositivity

Our statistical analysis confirms advancing age and male sex as associated factors of IgG seropositivity, as reported by other serosurveys [21,22,23,24]. In addition, we detected a significant interaction between age and sex for an IgM serostatus, indicating that males were increasingly likely to have IgM seropositivity with increasing age. We did not find statistical evidence for decreased seropositivity among individuals with international migration backgrounds, as similarly reported previously [21,22,24]. A Czech study [62] reported that immigrants rather sought residence in larger cities, entailing a lower mean risk of LB infection, as a potential explanation for comparably less prevalent tick-borne diseases among immigrants [22,27]. Since Hanover strongly varies in its green-space areas [69] and no accurate residence information was available for our analysis, we emphasize the importance of including information on outdoor activities and living areas, respectively, in future analyses.

We included the individual-level net equivalent income and education level for SES as controversially discussed risk factors. In our analysis, both factors were not associated with serostatus, as reported for the BGS98 and DEGS participants [23,70] or in a Norwegian study [28]. In contrast, recent findings from a highly educated urban German sample in Bonn found increased odds for seropositivity among highly educated individuals compared with intermediate education [24]. However, we identified the same trend for education in our data when interpreting the basic proportions (Table 1); no monotone relationship between seropositivity and income can be detected over the quartiles (Table 1). Additionally, among Slovenian Erythema migrans patients, high education was identified as an associated factor in addition to farm-related professions, as reported in other countries [30,71,72]. Then again, ecologic studies from the UK found a higher incidence of LB cases associated with higher regional socioeconomic indicators [26,27]. Congruently, a study in Munich, south Germany, demonstrated reduced green-space access in neighborhoods with below-average socioeconomic composition [73], characterizing its residents as potentially subject to lower exposure. We suspect the SES measurement method to have a particular influence on the reported findings.

Additionally, our findings indicate lower odds for seropositivity with an increasing BMI. Physical activity (PA), including outdoor activity, may be reduced among individuals with an elevated BMI [74,75,76], potentially relating to reduced PA-related tick exposure. However, our cross-sectional study design cannot assess the cause-and-effect direction. Similarly to BMI, we found an inverse relationship between increasing depressive symptomatology (PHQ-9) and positive serostatus. Again, individuals with depressive symptomatology may have been less exposed to ticks due to homestays [77].

### 4.4. Limitations

We identified limitations to our work. All included subjects were invited during 2014–2018; therefore, our reported seropositivity may not be utilized as a point estimate for a particular year. Furthermore, the NAKO participants were aged 20–69 at baseline and recruited from a somewhat urbanized area, potentially underestimating the regional seropositivity proportions. Since NAKO is a multi-themed study, the subjects were not asked for prior tick exposure, past diagnosis, or past/ongoing treatment for Lyme disease or corresponding symptoms. To our knowledge, we provided the FOI estimates for *B. burgdorferi* IgG-positivity for the first time and succeeded in achieving an additional view of the age distribution of seropositivity. However, future studies could compare more sophisticated FOI models, incorporating new seroreversion insights and exposure differences between population groups, e.g., sex-specific differences.

## 5. Conclusions

In conclusion, our work offers a recent baseline estimate of past human infection with *B. burgdorferi* s.l., complementing comprehensive tick abundance data for Hanover. The estimated local seropositivity is similar to previously aggregated information for the northern German federal states, indicating that a high local tick infection proportion does not necessarily result in elevated local seropositivity proportions. We found different proportions of seropositivity with age and sex (IgG and IgM), implying the need for risk communication for specific population groups, especially men and adults aged 50 and above. The relationship between BMI and depression and IgG serostatus suggests that healthier individuals are more likely to have past tick exposure, resulting in positive serostatus. Future investigations are required to unravel debated but potentially interwoven risk factors, i.e., socioeconomics, profession, outdoor activities, and tick exposure, to characterize the infection risk profile over age, including children and adolescents, for risk communication, and to monitor potentially changing climate-induced infection risk, resulting in a public health burden.

## Figures and Tables

**Figure 1 microorganisms-10-02286-f001:**
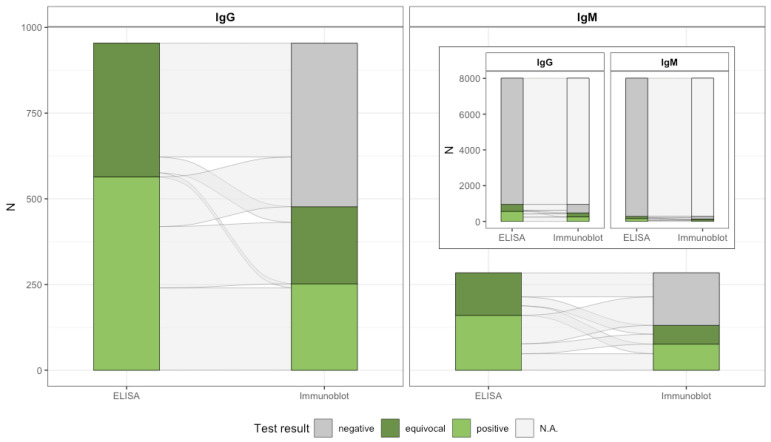
Alluvial diagram from two-tier sample testing for IgG and IgM antibodies against *B. burgdorferi* s.l.; IgG = Immunoglobulin G; IgM = Immunoglobulin M. We used ELISA as the screening test and a line immunoblot as the confirmatory test. ELISA-negative samples did not undergo confirmatory testing by protocol; therefore, subsequent immunoblot testing is not applicable (N.A.).

**Figure 2 microorganisms-10-02286-f002:**
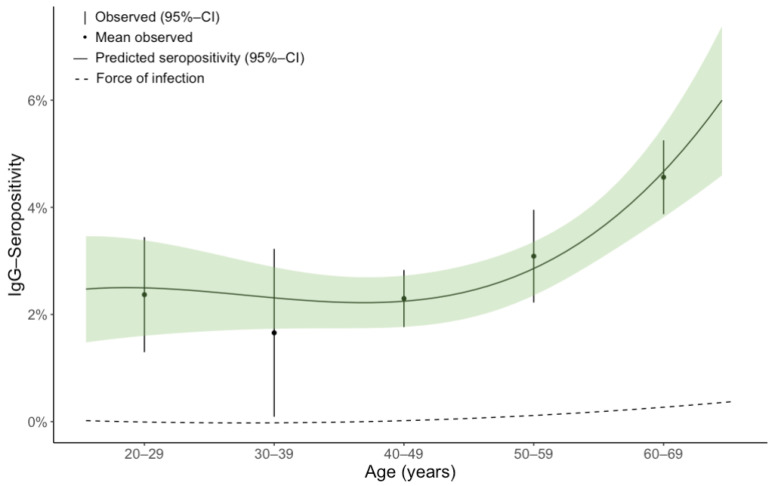
Observed IgG seropositivity for *B. burgdorferi* s.l. with the force of infection (FOI) and model-predicted seropositivity; IgG = Immunoglobulin G; observed seropositivity presented as error bars for 10-year age groups with mean value and 95% confidence interval; Grenfell and Anderson’s FOI [43] presented as a dashed line, representing the average change in the population’s seropositivity proportion. FOI model seropositivity predictions as a solid line with a 95% confidence interval (green).

**Figure 3 microorganisms-10-02286-f003:**
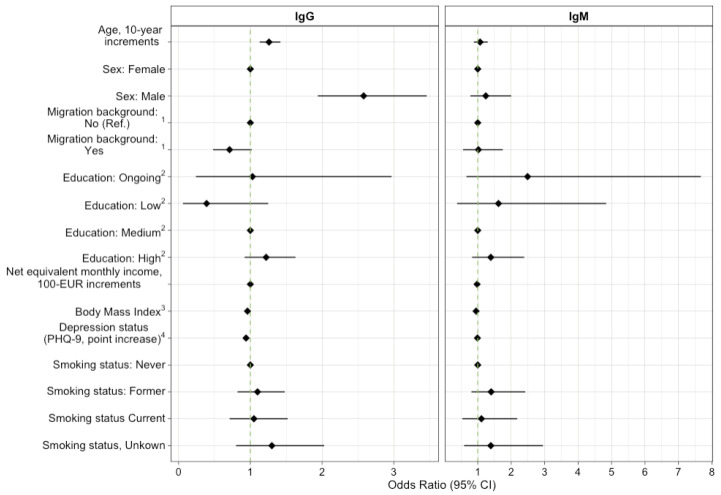
Odds ratios for IgG/IgM seropositivity from logistic regression. IgG = Immunoglobulin G; IgM = Immunoglobulin M; Ref. = reference; CI = confidence interval; Observations = 8009. We considered a sample as seropositive for Borrelia burgdorferi s.l. with positive or equivocal ELISA and subsequent positive immunoblot result (MiQ12) [33]; ^1^ Migration derived from a minimum set of indicators by Schenk et al. [44]; ^2^ Education level derived according to ISCED97 [45]; ^3^ BMI corresponding to the classification of the International Obesity Task Force [46]; ^4^ Depression status obtained from the 9-question Patient Health Questionnaire (PHQ-9.) [47].

**Table 1 microorganisms-10-02286-t001:** Population characteristics and crude seropositivity for *Borrelia burgdorferi* s.l.

Characteristics	Total (*N* = 8009)	IgG Seropositive (*n* = 252), Proportion (%, 95% CI)	IgM Seropositive (*n* = 76), Proportion (%, 95% CI)
**Age**			
20–29 years	831 (10.4%)	23/831 (2.8; 1.8–3.7)	11/831 (1.3; 0.7–2.0)
30–39 years	767 (9.6%)	14/767 (1.8; 1.0–2.6)	5/767 (0.7; 0.2–1.1)
40–49 years	2102 (26.2%)	47/2102 (2.2; 1.7–2.8)	22/2102 (1.0; 0.7–1.4)
50–59 years	2117 (26.4%)	67/2117 (3.2; 2.5–3.8)	17/2117 (0.8; 0.5–1.1)
60–69 years	1999 (25.0%)	91/1999 (4.6; 3.8–5.3)	19/1999 (1.0; 0.6–1.3)
70 years and older	193 (2.4%)	10/193 (5.2; 2.6–7.8)	2/193 (1.0; 0.0–2.2)
**Sex**			
Male	3991 (49.8%)	181/3991 (4.5; 4.0–5.1)	41/3991 (1.0; 0.8–1.3)
Female	4018 (50.2%)	71/4018 (1.8; 1.4–2.1)	35/4018 (0.9; 0.6–1.1)
**Migration Background ^1^**			
No	6389 (79.8%)	217/6389 (3.4; 3.0–3.8)	58/6389 (0.9; 0.7–1.1)
Yes	1616 (20.2%)	35/1616 (2.2; 1.6–2.8)	17/1616 (1.1; 0.6–1.5)
Missing	4 (0.1%)	0/4 (0.0; 0.0–0.0)	1/4 (25.9; 0.0–60.6)
**Education ^2^**			
Ongoing	172 (2.1%)	2/172 (1.2; 0.0–2.5)	3/172 (1.5; 0.1–3.4)
Low	203 (2.6%)	2/203 (1.0; 0.0–2.1)	3/203 (1.5; 0.1–2.9)
Medium	2680 (33.5%)	73/2680 (2.7; 2.2–3.2)	21/2680 (0.8; 0.5–1.1)
High	4480 (55.9%)	166/4480 (3.7; 3.2–4.2)	45/4480 (1.00; 0.8–1.2)
Missing	472 (6.9%)	9/472 (1.9; 0.9–2.9)	4/463 (0.9; 0.2–1.6)
**Net equivalent monthly income (Euro)**			
Median income (IQR)	2150 (1520–2917)	2150 (1633–3167)	1900 (1471–2533)
Quartile 1	1852 (23.1%)	49/1852 (2.6; 2.0–3.3)	18/1852 (1.0; 0.6–1.3)
Quartile 2	1972 (24.6%)	74/1972 (3.8; 3.0–4.5)	23/1972 (1.2; 0.8–1.6)
Quartile 3	1745 (21.8%)	46/1745 (2.6; 2.0–3.3)	18/1745 (1.0; 0.6–1.4)
Quartile 4	1813 (22.6%)	60/1813 (3.3; 2.6–4.0)	11/1813 (0.6;0.3–0.9)
Missing	621 (7.8%)	23/621 (3.7; 2.5–5.0)	6/621 (1.0; 0.3–1.6)
**Body Mass Index ^3^**			
Underweight	81 (1.0%)	1/81 (1.2; 0.0–3.3)	1/81 (1.2; 0.0–3.3)
Normal	3581 (44.7%)	128/3581 (3.6; 3.1–4.1)	37/3581 (1.0; 0.8–1.3)
Pre-obesity	2812 (35.1%)	78/2812 (2.8; 2.3–3.3)	30/2812 (1.1; 0.7–1.4)
Obesity class I	974 (12.2%)	29/974 (3.0; 2.1–3.9)	4/974 (0.4; 0.1–0.7)
Obesity class II	268 (3.3%)	9/268 (3.4; 1.5–5.2)	1/268 (0.4; 0.0–1.0)
Obesity class III	125 (1.6%)	2/125 (1.6; 0.0–3.4)	2/123 (1.6; 0.0–3.5)
Missing	168 (2.1%)	5/168 (3.0; 0.8–5.1)	1/168 (0.6; 0.0–1.6)
**Depression symptoms ^4^**			
None/minimal	4917 (61.4%)	183/4917 (3.7; 3.3–4.2)	49/4917 (1.0; 0.8–1.2)
Mild	1780 (22.2%)	32/1780 (1.8; 1.3–2.3)	12/1780 (0.7; 0.4–1.0)
Moderate	370 (4.6%)	6/370 (1.6; 0.5–2.7)	2/370 (0.5; 0.0–1.2)
Moderately severe	134 (1.7%)	2/134 (1.5; 0.0–3.2)	2/134 (1.5; 0.0–3.2)
Severe	36 (0.4%)	0/36 (0.0; 0.0–0.0)	0/36 (0.0; 0.0–0.0)
Missing	772 (9.6%)	29/772 (3.8; 2.6–4.9)	11/772 (1.4; 0.7–2.1)
**Smoking status**			
Never	1695 (21.2%)	48/1695 (2.8; 2.2–3.5)	17/1695 (1.0; 0.6–1.4)
Former	1396 (17.4%)	49/1396 (3.5; 2.7–4.3)	11/1396 (0.8; 0.4–1.2)
Current	694 (8.7%)	23/694 (3.3; 2.2–4.4)	8/694 (1.2; 0.5–1.8)
Unknown	393 (4.9%)	15/393 (3.8; 2.2–5.4)	7/393 (1.8; 0.7–2.9)
Missing	3831 (47.8%)	117/3831 (3.1; 2.6–3.5)	33/3831 (0.9; 0.6–1.1)

IgG = Immunoglobulin G; IgM = Immunoglobulin M. We considered a sample as seropositive for *Borrelia burgdorferi* s.l. with positive or equivocal ELISA and subsequent positive immunoblot results (MiQ12) [33]; ^1^ Migration derived from a minimum set of indicators by Schenk et al. [44]; ^2^ Education level derived according to ISCED97 [45]; ^3^ BMI corresponding to the classification of the International Obesity Task Force [46]; ^4^ Depression symptoms classification obtained from the 9-question Patient Health Questionnaire (PHQ-9) [47].

**Table 2 microorganisms-10-02286-t002:** Total crude and weighted seropositivity for *Borrelia burgdorferi* s.l. by three seropositivity algorithms.

Antibody Type	Seropositivity Definition	Crude Numbers	Crude % (95% CI)	Weighted Estimate % (95% CI) ^1^
IgG	ELISA: positive or equivocal and line blot: positive (MiQ12) ^2^	252/8009	3.1 (2.8–3.5)	3.0 (2.7–3.4)
	ELISA: positive and line blot: positive or equivocal or ELISA: equivocal and line blot: positive	431/8009	5.4 (4.9–5.9)	5.2 (4.7–5.7)
	ELISA: positive ^3^	564/8009	7.0 (6.5–7.6)	6.8 (6.3–7.4)
IgM	ELISA: positive or equivocal and line blot: positive (MiQ12) ^2^	76/8009	0.9 (0.7–1.2)	0.9 (0.7–1.2)
	ELISA: positive and line blot: positive or equivocal or ELISA: equivocal and line blot: positive	105/8009	1.3 (1.1–1.6)	1.4 (1.2–1.7)
	ELISA: positive ^3^	160/8009	2.0 (1.7–2.3)	2.1 (1.8–2.4)

IgG = Immunoglobulin G; IgM = Immunoglobulin M; ELISA = enzyme-linked immunosorbent assay; ^1^ Seropositivity weighted by the age and sex ratio of our target population (Hanover) based on the 2020 update of the 2011 census (www.detatis.de) [36] to approximate the seropositivity for the general population; ^2^ Seropositivity according to the MiQ12 standard [33]; ^3^ Only positive ELISA considered regardless of line blot result.

## Data Availability

Data of the NAKO are generally not available to the public due to strict data protection regulations. However, scientists can apply for data use according to the official usage regulation specifications. Please refer to https://transfer.nako.de for further information.

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
