# Peer review of "Seroepidemiology of Borrelia burgdorferi s.l. among German National Cohort (NAKO) Participants, Hanover"

_microorganisms, 2022, doi:10.3390/microorganisms10112286_

Round 1

Reviewer 1 Report

The manuscript by Hassentein et al., investigates the serum from a large cohort of patients for antibodies against Lyme disease in Hanover. The sampling is large and the data is certainly relevant even if 4 years after the last collection of samples.

1. In the Materials and Methods section- there is no description of how the blood was collected, obtaining of serum and storage of the serum and how long after storage were the samples tested.

2. I could not find anywhere in the results section if any of the patients that tested positive for IgM also were positive of IgG.

3. I understand that one of the measures that patients that are part of NAKO are asked concerns depression and smoking status. I do not find this to be relevant to this study and suggest the authors remove this from Table 1 and their discussion/conclusion as the correlation is very superficial.  Ideally the authors should have looked at symptoms/ailments that are usually associated with Lyme Disease or Chronic Lyme Disease including joint pain, aches, constant headaches, etc.

4. The authors need to state in their limitations that the patients were never asked if they had prior tick exposure or had been diagnosed/treated previously for Lyme Disease.

Author Response

Dear Reviewer, thank you for your efforts. Please find our answers in the attached document. Kind regards.

Reviewer 2 Report

The paper by Hassenstein and colleagues is a nice piece of work describing a local multiannual serosurvey for Borrelia burgdorferi s.l. infection in a sample of the population of  Hanover, Lower Saxony, Germany.

The authors identify advanced age and male sex as risk factors for B. burgdorferi seropositivity, and they find and discuss particular lifestyle behaviours that may reduce or increase tick bite exposition influencing the serological status.

The manuscript is well written, the background and aims are appropriately explained, the methods are adequate, and the results are clearly exposed and thoroughly discussed, drawing well supported conclusions.

Accordingly, I have no major criticisms to do but some formal issues that I have observed, for instance:

Line 24-25: better “examined from 2014 to 2918”

Line 79: better “past infections are reflected”

Line 223: “Figures S1, S2” instead “Supplementary Figures 2, 3”

Line 262: “Table S1” instead “Supplementary Table 1”

Author Response

(The authors gave the same response as above.)

Reviewer 3 Report

The study performed by Hassanstein et al., assessed the seroprevalence of Borrelia burgdorferi in Hanover. Overall, the study is quite well-performed as per the sample size and analysis. However, this reviewer finds a lack of novelty in the study and hence, there are several gaps to be filled. In addition, the manuscript needs extensive editing in the English language.

1. Abstract needs rephrasing.

2. Introduction: There are several published studies focusing on the seroepidemiology of B. burgdorferi in Hanover. The introduction should include some statements stating why this study is different than others and how this study adds novelty to already existing knowledge.

3. Methods: Well-explained

4. Results: Authors are suggested to plot the graph for the data in Table-4. This will help the readers to understand more.

Author Response

(The authors gave the same response as above.)

Round 2

Reviewer 1 Report

The authors have been able to address my concerns and suggestions and I am satisfied with the corrected version of this manuscript . 

Reviewer 3 Report

In the revised manuscript submitted by Hassanstein MJ. et al., authors have included the suggested changes and the manuscript can be accepted in the present form.